# Role of root exudates on assimilation of phosphorus in young and old *Arabidopsis thaliana* plants

Hugo A. Pantigoso[1], Jun Yuan[2], Yanhui He[1,3], Qinggang Guo[1,4], Charlie Vollmer[5], Jorge M. Vivanco[1]*

1 Department of Horticulture and Landscape Architecture, Center for Rhizosphere Biology, Colorado State University, Fort Collins, Colorado, United States of America, 2 Jiangsu Provincial Key Lab of Organic Solid Waste Utilization, Jiangsu Collaborative Innovation Center for Organic Solid Waste Utilization, Nanjing Agricultural University, Nanjing, China, 3 Key laboratory for Green Processing of Chemical Engineering of Xinjiang Bingtuan, School of Chemistry and Chemical Engineering, Shihezi University, Shihezi, China, 4 Institute of Plant Protection, Hebei Academy of Agricultural and Forestry Science, Baoding, China, 5 Department of Statistics, Colorado State University, Fort Collins, Colorado, United States of America

* J.Vivanco@colostate.edu

**Data Availability Statement:** All relevant data are within the paper and its Supporting Information files.

## Abstract

The role of root exudates has long been recognized for its potential to improve nutrient use efficiency in cropping systems. However, studies addressing the variability of root exudates involved in phosphorus solubilization across plant developmental stages remain scarce. Here, we grew *Arabidopsis thaliana* seedlings in sterile liquid culture with a low, medium, or high concentration of phosphate and measured the composition of the root exudate at seedling, vegetative, and bolting stages. The exudates changed in response to the incremental addition of phosphorus, starting from the vegetative stage. Specific metabolites decreased in relation to phosphate concentration supplementation at specific stages of development. Some of those metabolites were tested for their phosphate solubilizing activity, and 3-hydroxypropionic acid, malic acid, and nicotinic acid were able to solubilize calcium phosphate from both solid and liquid media. In summary, our data suggest that plants can release distinct compounds to deal with phosphorus deficiency needs influenced by the phosphorus nutritional status at varying developmental stages.

## Introduction

Phosphorus is an essential element for plant growth and development [1], and a non-renewable resource [2,3]. Despite the fact that the total amount of phosphorus is high in most agricultural soils, crop yields are often limited by low availability due to the non-soluble form and low mobility of this nutrient [4,5]. It has been estimated that residual phosphorus fertilizer known as 'phosphorus legacy' in soil can be sufficient to sustain crop yield for the next century and could alleviate expected phosphorus shortages in the next 50 years [6]. Hence, studies addressing potential solutions to exploit soil phosphorus reserves are needed.

**Funding:** J. Y. was supported by National Postdoctoral Program for Innovative Talents (BX201600075) and Natural Science Foundation of Jiangsu Province (BK20170724), Y. H. was supported by China Scholarship Council (No. 201709505007). The funders had no role in study design, data collection and analysis, decision to publish, or preparation of the manuscript.

Plants have developed several strategies for acquisition of phosphorus in low nutrient environments mainly by modifying root structure and changing the soil chemical properties in the rhizosphere [7]. These mechanisms include longer root formation and increases of root:shoot ratio allowing the transport of phosphorus from the roots to the shoots [8,9]. Certain plants such as *Lupinus* sp. can promote the formation of cluster roots to secrete phosphorus solubilizers such as citrate and malate in sufficient quantities to lower the rhizosphere pH, thus enhancing the movement of phosphorus and consequently plant uptake [10,11]. The secretion of phosphorus solubilizers is not restricted to cluster root-forming plants. Several other species such as alfalfa, spinach and radish have also been documented to increase the efflux of organic anions as a result of a lack of phosphorus available in the soil [12–14].

Increasing phosphorus solubility can also be achieved by modifying the rhizosphere chemistry. Root-secreted phosphorus solubilizers are capable of increasing solubility of a variety of insoluble phosphorus forms in the soil such as organic phosphorus, inorganic phosphorus like calcium phosphate, and humic substances bonded to phosphorus anions such as Al (III) and Fe (III). They can be classified as protons or $OH^-/HCO_3^-$ equivalents, redox equivalents and di- and tricarboxylic acid anions [13]. The mechanisms used by plants to solubilize phosphorous vary according to the plant species, nutritional status of the plant, and soil and environmental conditions [15]. However, organic acids such as oxalate, citrate, and malate are recurrent in a variety of plant species and thereby are the most studied means used by plants to solubilize phosphorous [13]. Considerably less research has been performed to explore the total root exudate profile and to identify other compounds exerting similar and complementary functions in the rhizosphere.

All plants share a similar need for phosphorus, but this need differs broadly based on the crop type and its developmental stage [16,17]. In general, most crops require early phosphorus supplementation for optimum yield [18]. Nevertheless, higher amounts of phosphorus in later growth stages are required proportional to increases in the biomass of the plant [19–22]. Due to the variation of phosphorus demand during the plant's lifetime, it becomes necessary to fully understand fluctuations of root exudates as a means to solubilize the phosphorus present in the of the soil.

In this study, we tested the effect of three phosphate fertilization levels on root exudate composition of *A. thaliana* at distinct developmental stages. We hypothesized that phosphate status will promote a shift in the relative concentrations of certain root exudates with inorganic phosphate solubilizing properties and certain metabolites involved in phosphate solubilization will be inhibited under high phosphate concentrations. The results showed that the total exudates changed in response to the addition of phosphate, and that certain metabolites were reduced under increasing phosphorus amendments at varying growth points. As a proof of concept, some of the metabolites that decreased in quantity to the phosphorus addition were tested and four of them were found to solubilize phosphate.

## Materials and methods

### Plant growth conditions, phosphorus fertilization and collection of root exudates

*Arabidopsis thaliana* (L.) Heynh wild-type Columbia seeds were obtained from Lehle Seeds (Round Rock, USA) and surface sterilized with Clorox bleach (Sodium Hypochlorite, 8.25%) for 1 min. Seeds were rinsed with distilled water 3 times and plated on different phosphorus levels of Murashige and Skoog (MS) agar (1.5%) (supplemented with 3% sucrose) in square plates. The plates were placed vertically in a growth chamber (Percival Scientific) at 25 ± 2 ˚C with a photoperiod of 16 h: 8 h, light: dark for germination. The germinated seedlings were

grown in three phosphate levels: full strength (100%, 1.25 mM), half (50%, 0.625 mM) and a quarter (25%, 0.3125 mM) in solid MS medium as described above. The three phosphate levels used in this study did not stimulate the plant starvation response, which is generally activated at values below 2 μM phosphorus in the soil solution [23]. Instead, we followed a low and high fertilization regime used commercially in agriculture [24]. phosphate concentration was adjusted with $Na_2HPO_4$ and $NaH_2PO_4$, and phosphate -free MS medium was used as basal medium. After seven days of growth, the seedlings were transferred to six well plates, each well containing 5 mL of liquid MS with 1% sucrose with one individual seedling per well containing distinct phosphate levels as described above. The seedlings grown in solid MS at the particular phosphate level were placed under the same phosphorus level under liquid M conditions. All the plates were placed on an orbital shaker at 90 rpm under 25 ± 2 ˚C and lit by cool white fluorescent light (45 μmol $m^{-2}$ $s^{-1}$) with a photoperiod of 16 h: 8 h, light: dark. The nutrient solutions were replaced every week by transferring the plants to new six well plates with 5 ml of fresh liquid MS medium under the same phosphate levels as stated above.

Root exudates were collected as follows: 1–3 days after transplanting for seedling stage, 8–10 days after transplanting for vegetative stage, and 15–17 days after transplanting for bolting stage. Prior to the collection of root exudates, plants were removed and washed mildly with sterile water and subsequently placed in new wells containing 5 ml of sterile water for two days. We used sterile distilled water to prevent the interference of exogenously supplemented salts and sucrose present in the Murashige and Skoog media in subsequent GC-MS analyses of the root exudates. The solution in which the plants were floating was collected and considered as the root exudate. The solution of one plate containing six wells with six individual plants was pooled and considered as one replicate. We used three replicates for each stage under three phosphorus levels. The root exudates were filtered through a 0.45 μm (Millipore, MA) to remove root sheathing and root border-like cells and were freeze-dried and stored at -20 ˚C for further analyses.

## Gas Chromatography-Mass Spectrometry (GC-MS) of root exudates

To characterize the chemical composition, root exudates were processed as described by Chaparro et al, [25] and subjected to gas chromatography—mass spectrometry GC-MS analyses at the Genome Center Core Services, University of California Davis. Extracts were briefly dried under nitrogen gas and then methoximated and trimethylsilylated. The derivatives were analyzed by an Agilent 6890 gas chromatograph (Santa Clara, CA) containing a 30-m-long, 0.25-mm inner diameter rtx5Sil-MS column with an additional 10 m integrated guard column. Metabolites were detected using the BinBase algorithm [26] and identified by comparing the retention index and mass spectrum of each analyte against the Fiehn mass spectral library from the West Coast Metabolomics Center, University of California Davis.

## Statistical analysis of total root exudate data

To discover the differential expression levels of compounds across the plant's growth stages and fertilizer levels, R statistical software (R Core Team, 2017) was used to perform principal components analysis (PCA) on all annotated compounds. For each of the following analyses, we performed a PCA by first separating the total root exudate data by plant growth stage. Within each plant growth stage (e.g. bolting), we performed a centered and scaled PCA. By verifying that the first two principal components explained a sufficient amount of the variance in the compound(s) expression levels, we were able to determine which compounds had the highest correlation with these components through the magnitude of the variance. The variances are representative of a compound at each growth stage and at a particular phosphate

level. The largest variance represents the highest correlation to the principal components. This method allowed us to determine which compounds explained most of the variance across the fertilizer levels and for each of the plant's growth stages.

Significant differences between phosphorus amendment and compound counts level per developmental stage were analyzed using a one-way ANOVA. Tukey HSD test was used to identify significance ($p < 0.05$) among phosphorus treatments.

## Qualitative analysis of phosphate solubilizing ability of compounds derived from root exudates

From a list of selected compounds, only 13 of them were diluted in $ddH_2O$ at the desired concentration (100 mM) (Table 1). We used the reported concentration of organic acids in the rhizosphere (1 μM to 100 mM) as reference to select the concentration of the compounds tested in this study [27–29]. These 13 compounds were qualitatively evaluated for their phosphate solubilizing abilities on National Botanical Research Institute phosphate growth medium (NBRIP) solid medium containing: 10.0 g glucose, 5.0 g $Ca_3(PO_4)_2$, 0.2 g NaCl, 0.5 g $MgSO_4 \cdot 7H_2O$, 0.5 g $(NH_4)_2SO_4$, 0.2 g KCl, 0.03 g $MnSO_4$, 0.003 g $FeSO_4 \cdot 7H_2O$, 12 g agarin 1 L

**Table 1. Five top compounds from root exudates identified by Principal Component Analysis (PCA).**

| Growth Stage | P level (%) | Compound name | |
|---|---|---|---|
| Seedling | N/A | N/A | |
| Vegetative | 25 | Nicotinic acid | * |
| | | 4-hydroxybutyric acid | * |
| | | 3-hydroxypropionic acid | * |
| | | 1-monostearin | |
| | | 1-monopalmitin | |
| Vegetative | 50 and 100 | Threonine | * |
| | | Proline | * |
| | | O-acetylserine | * |
| | | Leucine | * |
| | | Alanine | * |
| Bolting | 25 | Threonic acid | |
| | | Octadecanol | |
| | | Malic acid | * |
| | | Glycine | * |
| | | Galactinol | * |
| Bolting | 50 | Scopeletin | |
| | | phenylacetamine | |
| | | 5-aminovaleric acid | * |
| | | 1-monopalmitin | |
| | | 1-monoheptadecanoyl glyceride NISTT | |
| Bolting | 100 | Sophorose | |
| | | Guanine | |
| | | Glutamic acid | * |
| | | Adenine | |
| | | 1-Kestose | |

Compounds are divided by phosphate level and plant developmental stages. Compounds diluted in ddH2O at 100 mM are indicated (*).

water, pH: 7.0–8.0 [30]. All compounds used in this experiment were purchased from Thermo-Fisher Scientific. The specific solution (100 μL) of each compound was placed on NBRIP solid medium. The plates were inoculated at room temperature and let to sit overnight. Phosphate solubilizing ability was visually judged as a clear halo around every drop of solution containing the given compound. Briefly, the test of the relative efficiency of isolated metabolites was carried out by selecting the metabolites that were capable of producing a halo or clear zone in the surrounding medium by the dissociation of inorganic minerals such as calcium phosphate.

## Quantitative analysis of phosphorus solubilizing ability of water-soluble compounds (individually and combined)

For quantification, 35 μL of the same concentration previously tested (100 mM) of the 13 compounds were added to 5 mL liquid NBRIP medium resulting in a final concentration of 7mM. The tubes were then placed at a continuous agitation at 150 rpm on a rotary shaker for 72 h. Afterwards, the solution was centrifuged at 6000 rpm for 5 min, and the supernatant was filtered with 0.2 μm filter (Thermo-Fisher Scientific). Liquid NBRIP medium without compound addition was used as control. The concentration of phosphorus in the supernatant was analyzed according to with the protocol of Soltanpour et al. [31] and measured by means of inductive coupled plasma-optical emission spectrometer (ICP-OES) (Perkin Elmer 7300DV) at the Soil, Water and Plant Testing Laboratory of Colorado State University. This experiment and analysis were repeated twice with 3 repetitions.

In order to determine the potential cumulative effect of 3-hydroxypropionic acid, malic acid, nicotinic acid, and glutamic acid, they were combined and the available phosphorus (mg $l^{-1}$) was determined by OES-ICP. A compound mixture that included the previously tested concentration (7 mM per compound) was assayed in order to compare if the combination of compounds would equal or surpass the effect of a single compound. Briefly, 35 μL (100 mM) of each compound was added to 5 mL liquid NBRIP medium resulting in a final concentration of 7mM. Each compound added to the pool had a concentration of 7 mM. Thus, the combination effect of four compounds were tested in a liquid NBRIP medium.

## Statistical analysis for quantitative phosphate release

Significant differences between phosphorus content measured in NBRIP liquid media was analyzed using a one-way ANOVA. Tukey HSD test was used to identify significant treatments in multiple comparisons. P-value were considered significant below 0.05. Assumptions of normality and homogeneity of the data were confirmed prior to the analysis.

## Results

### Effect of phosphate levels on *A. thaliana* root exudates at different plant developmental stages

The effect of increasing phosphate at three concentrations on the root exudates of *A. thaliana* was analyzed at various developmental stages. In total, 456 compounds were detected by GC-MS among the treatments. The data set was reduced to 201 annotated compounds, and only these were kept for statistical analysis. The grouping of the compounds in the plot maintained the same pattern even after subtracting the non-annotated compounds (S1 Fig), suggesting that all of the differentially expressed compounds were indeed annotated by the GC-MS analyses.

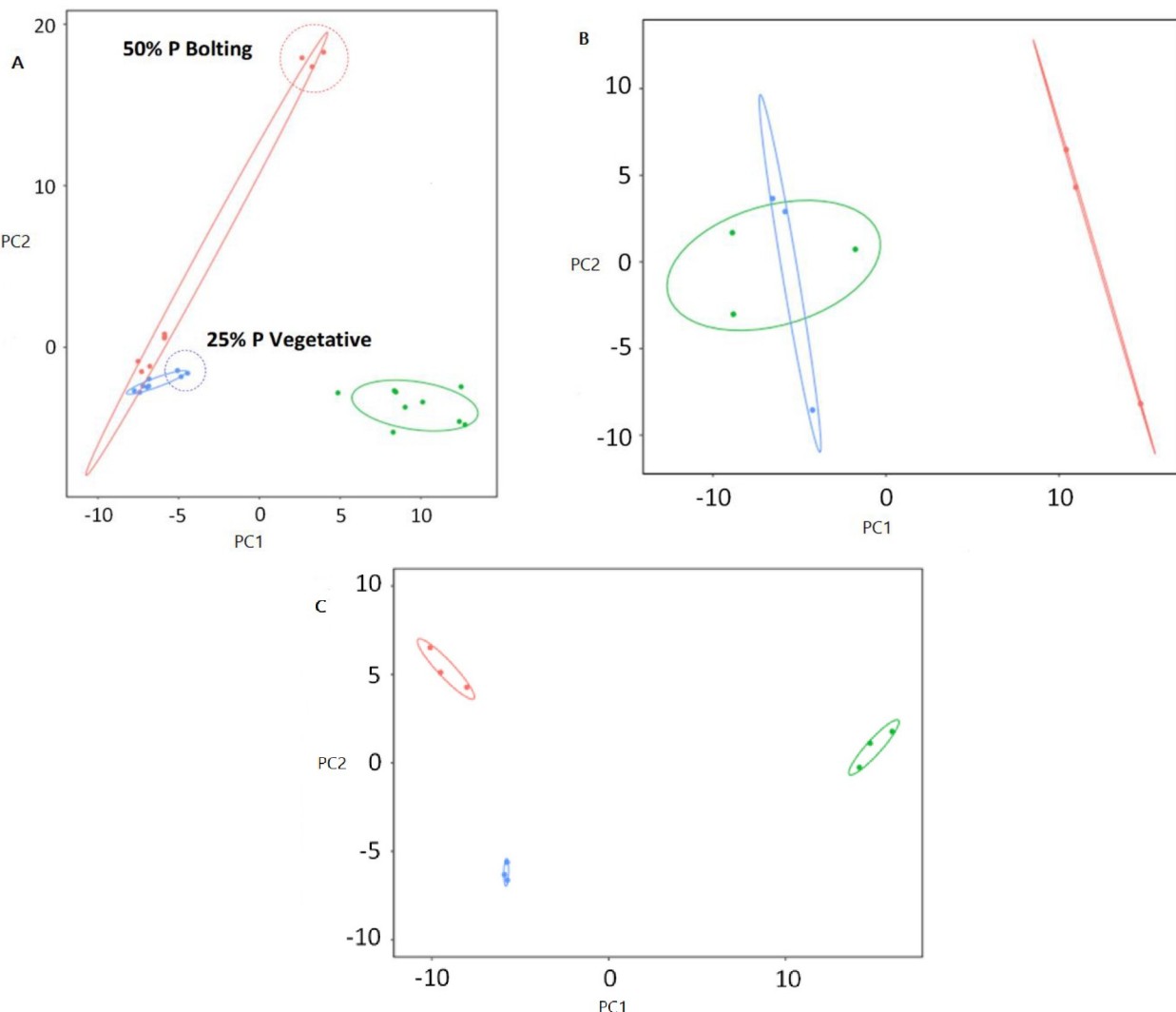

**Fig 1. Root exudate compounds diverge in response to plant developmental stage and fertilization rate. (A)** 201 annotated compounds with proper identification detected using GC-MS was analyzed using a Principal Component Analysis (PCA) graph. PCA show dissimilarity of metabolite expression profiles between plant growth stages where PC1 explained 29.8 and PC2 21.7% of the variability. All phosphorus levels (25%, 50% and 100%) are present in each of the plant growth stages shown; seedling (green), vegetative (blue), and bolting (red). Dotted circle highlight clusters of particular fertilization levels. **(B)** Plot of PCA for vegetative stage only where PC1 explained 43.6% and PC2 14.6% of the variability. Compounds grouped by phosphorus treatments: 25% P (red) fertilization clusters separates from 50% P (blue) and 100% P (green). **(C)** Plot of PCA for bolting stage only where PC1 explained 64.3% and PC2 12.9% of the variability. Compounds grouped by phosphorus treatments: 50% phosphorus (green) separated from 25% phosphorus (red) and 100% phosphorus.

The variability in our data, after subtracting the non-annotated compounds, was analyzed using a principal component analysis (PCA) where variability of component 1 (PC1) accounted for 29.8%, while component 2 (PC2) accounted for 21.7%. We determined that the plant's developmental stage was responsible for the separation of the compounds in three marked groups: seedling, vegetative and bolting (Fig 1A) as previously reported [25,32,33]. When analysis included fertilizer levels segregation was observed in certain developmental stages and phosphate levels (S2 Fig). Overall, phosphate levels did not cause a significant separation on the root exudate patterns at the seedling stage, but the separation was observed in

the vegetative and particularly in the bolting phases (Fig 1A). In the vegetative stage, a clear parting was observed at 25% phosphate compared to the 50 and 100% treatments (Fig 1B). In the bolting stage, there was a clear division between the three fertilizer levels, but the 50% level was the most distant rate (Fig 1C).

## Differences in compound-levels in the vegetative and bolting stages due to phosphate fertilization

A separate analysis was performed to determine correlations between different and highly expressed compounds for a specific fertilization level and developmental stage. We focused our analysis on just the 25% phosphate at the vegetative stage, and all three treatments at the bolting stage because these treatments had the highest dissimilarity in the PCA. In addition, the 50% and 100% treatment at the vegetative stage were grouped because they were clustered in the PCA. For each of these treatments, we found the five top compounds that explained the largest proportion of the variance in the principal components (Table 1). In addition, the abundance of the compounds based on phosphate level and growth stage were determined (S3 and S4 Figs). We found some interesting patterns, such as that some compounds decreased expression upon increased fertilization (e.g. 3-hydroxypropionic acid, malic acid, galactinol), while other compounds showed a positive correlation with phosphate amendment (e.g. guanine, glutamic acid, sophorose).

## Root-exudate metabolites solubilize calcium phosphate in solid and liquid media

We then tested the ability of some of the selected compounds irrespective of their abundance upon fertilization to solubilize phosphate solubilization. We found that 3-hydroxypropionic acid, malic acid, and nicotinic acid formed a clear halo when tested at a concentration of 100 mM, indicating the ability of this compound to release free phosphate from calcium phosphate (Fig 2). At 100 mM, none of the remaining tested compounds solubilized phosphate detectably.

The phosphate-solubilizing effect of the selected compounds was further tested using a more sensitive technique (OES-ICP) where the compounds where tested at a final concentration of 7 mM in liquid NBRIP medium. The results showed that in addition to 3-hydroxypropionic acid, malic acid, and nicotinic acid, glutamic acid also had phosphate solubilizing ability. Using this method, glutamate, malate, and nicotinic acid solubilized approximately ten

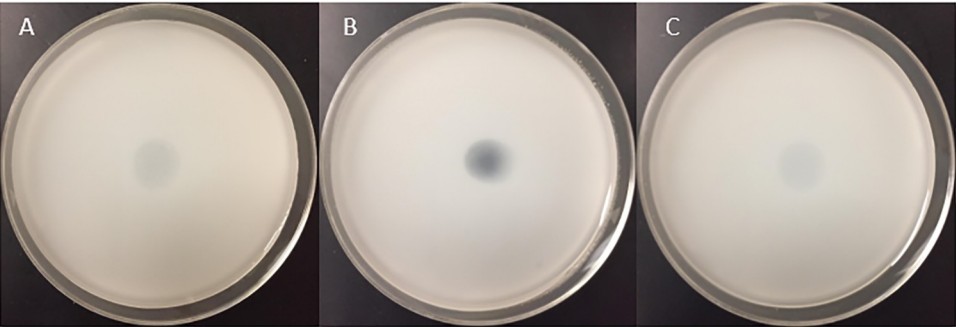

**Fig 2. Qualitative phosphate-solubilization analysis of compounds using a calcium-phosphate based medium (NBRIP). (A)** 3-Hydroxypropionic acid **(B)** malic acid and **(C)** nicotinic acid.

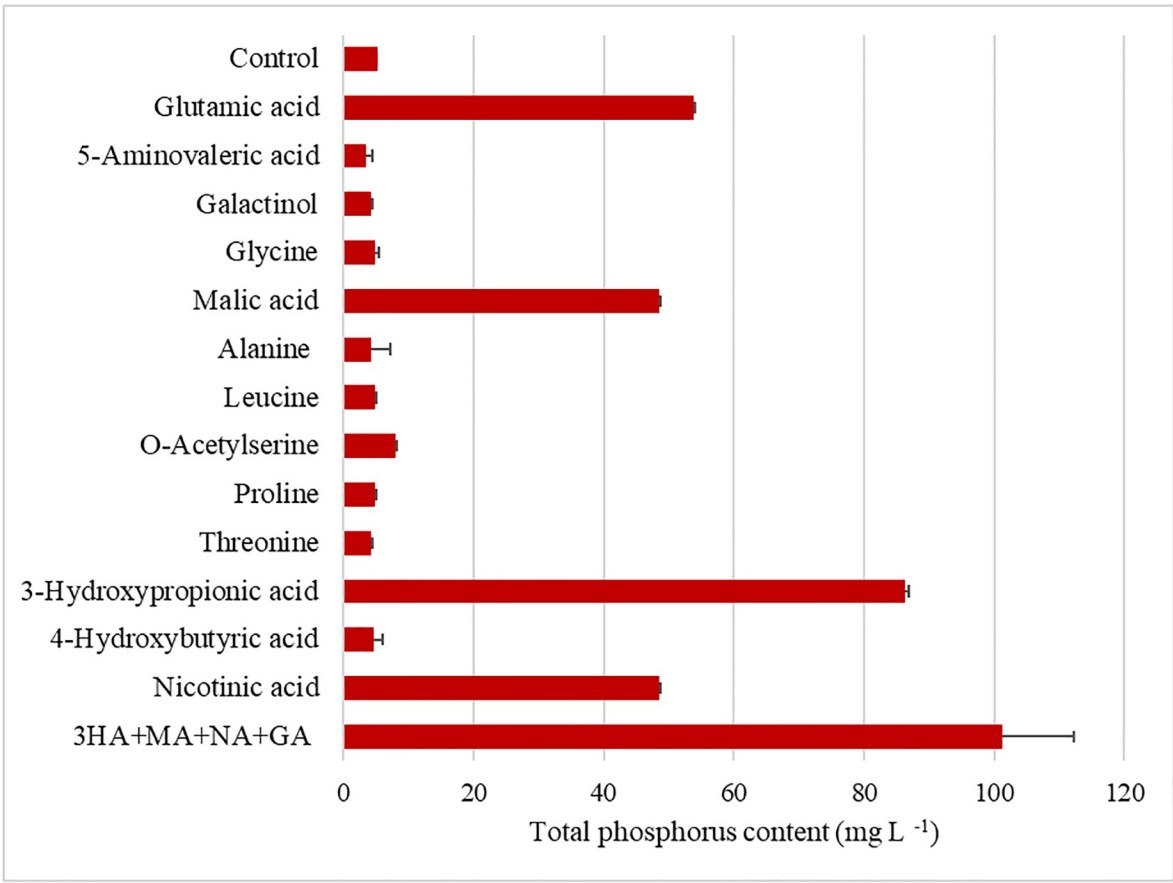

**Fig 3. Quantitative phosphate-solubilization analyzed by coupled plasma-Optical Emission Spectrometer (ICP-OES) in 13 identified compounds.** Available phosphorus content in NBRIP liquid media after incubation of 72 hours with each of the 13 compounds at 7mM concentration. 3HA+MA+NA+GA treatments is the combination of 3-hydroxypropionic acid (3HA), malic acid (MA), nicotinic acid (NA) and glutamic acid (GA). Each compound added to the pool had 1.75 mM, 7 mM and 28 mM concentration.

times the amount of phosphorus present in the control (5.34 mg L$^{-1}$) whereas 3-Hydroxypropionic acid solubilized almost fifteen times more (Fig 3).

Further analysis aimed to test the combined effect of all the four compounds on NBRIP medium at 7 mM concentration of each compound resulting in a twenty times higher available phosphate (101.21 mg L$^{-1}$) compared to the control (Fig 3).

## The plant mediates changes in secretion of solubilizing compounds in response to phosphorus status and developmental stage

We further analyzed the exudation level upon phosphate fertilization of 3-hydroxypropionic acid, malic acid, nicotinic acid, and glutamic acid in the different developmental stages. Depending on the compound, our results showed that the cumulative secretion levels of the compounds increased, decreased or remained statistically similar ($p > 0.05$) as a function of phosphate amendment (Fig 4). At the seedling stage, changes in cumulative secretion of the four solubilizers was not related to phosphate level significantly which agrees with the PCA analysis (Fig 1A). Interestingly, at the vegetative stage, nicotinic acid and 3-hydroxypropionic acid showed higher abundance at the lowest phosphorus level (0.3125 mM) and decreased

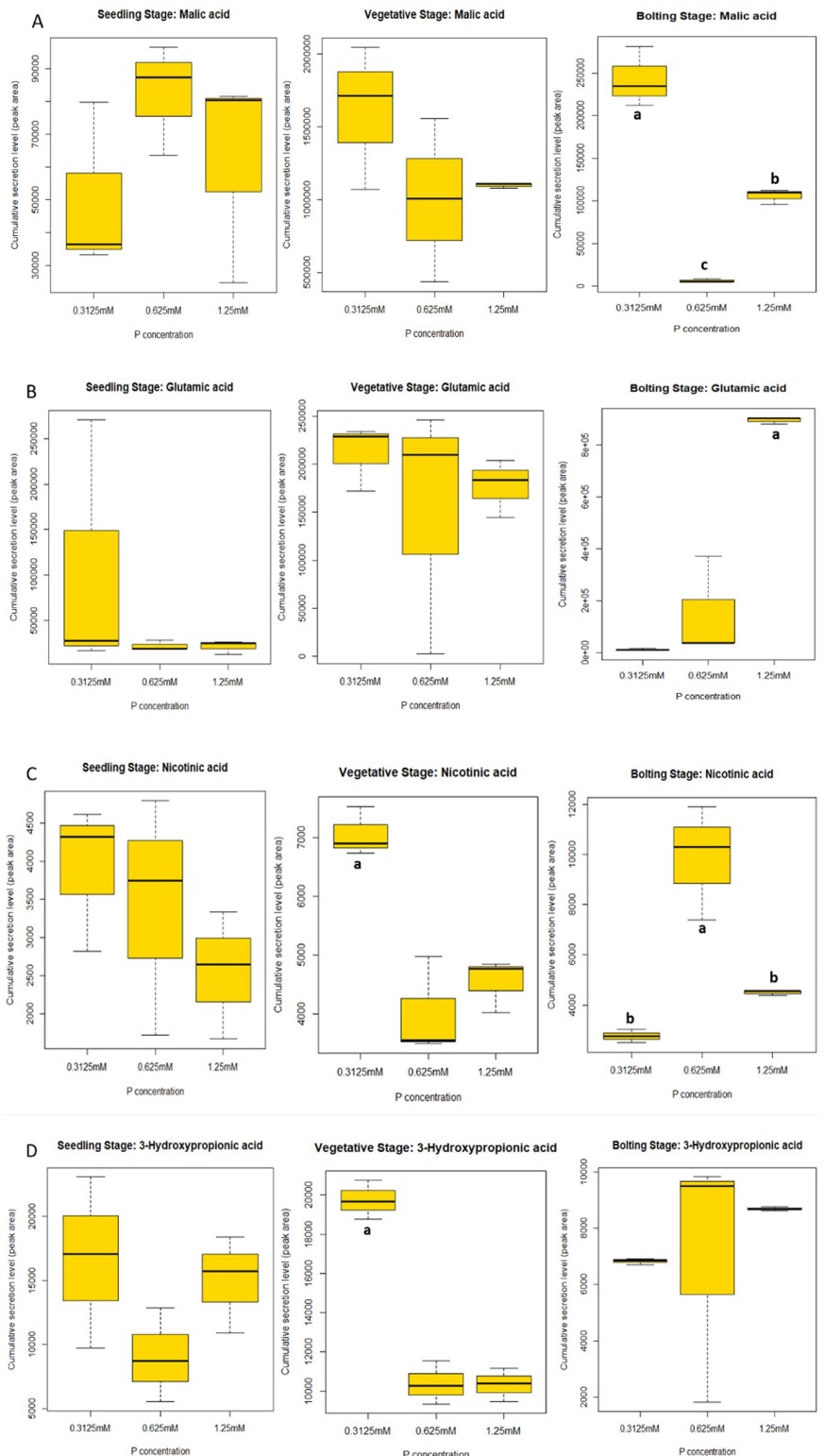

**Fig 4. Phosphate-solubilizer compounds showing changes in cumulative root secretion levels at three distinct developmental stages (p<0.05) in response to increasing phosphate addition (0.312, 0.625 and 1.25 mM).** Malic acid (**A**), Glutamic acid (**B**), Nicotinic acid (**C**), 3-hydroxypropionic acid (**D**).

significantly ($p<0.05$) for the two higher levels (0.625 and 1.25 mM). Malic and glutamic acid followed a similar pattern; however, their changes were not statistically significant. At the bolting stage, differences of compound cumulative secretion were observed for glutamic acid, malic acid and nicotinic acid, but not for 3-hydroxypropionic acid which did not increase or decrease following an incremental phosphorus level. Cumulative secretion of malic acid was reduced significantly from 0.3125 to 0.625 mM phosphate treatments, and then incremented its cumulative secretion for the highest treatment (1.25 mM). However, secretion levels of malic acid for the two highest phosphate treatments (0.625 and 1.25 mM) were below the value for the lowest rate (0.3125 mM) (Fig 4A). Similarly, cumulative secretion of nicotinic acid increased significantly from 0.3125 to 0.625 mM but dropped for 1.25 mM of phosphate (Fig 4C). Lastly, glutamic acid consistently increased upon higher phosphorus fertilization reaching its peak at 1.25 mM of phosphate (Fig 4B).

## Discussion

For the most part, the influence of phosphorus fertilization on root secretion has been studied at specific developmental stages [34,35], among genotypes of the same plant species [36] or between different species [37]. For instance, Vengavasi et al. [36] found cultivar-dependent differences in root exudation of soybeans grown under phosphorus -sufficient versus phosphorus -deficient conditions. It is worth noting that these plants were sampled at the reproductive stage of growth, a metabolically active stage with higher demand for energy and phosphorus nutrition. In a different study, the authors found differences in root exudation in maize seedlings grown in phosphorus -sufficient and phosphorus -deficient conditions [34]. In contrast, here we studied the interplay of increasing phosphorus fertilization on root exudation at different plant developmental stages (seedling, vegetative and bolting) in *A. thaliana*.

Our results showed that the root exudate profiles were similar within the seedling stage across all phosphate fertilization treatments (Fig 1A). Thus, we hypothesized that at this growth stage roots did not respond to phosphorus fertilization. *A. thaliana* is considered a plant that can thrive in marginal soils where optimum nutrient conditions are limited [38], and at early developmental stages the plant does not require high amounts of phosphorus as a mechanism to cope with poor soil conditions [39]. In fact, it has been shown that the reserves of phosphorus in the seeds of several plants, including *Brassicaceae* species, can support seedling growth in a medium lacking phosphorus for at least four weeks after germination [40–42,39].

In contrast, maize, a monocotyledonous plant, possesses two genes induced by phosphate starvation in its genomes compared to five in eudicots such as *A. thaliana* [43,44]; suggesting a difference in phosphate responsiveness between these two plant groups. In addition, signs of phosphorus scarcity in eudicots (e.g. *A. thaliana*) is often observed at later stages of growth [45]. In the vegetative stage, the root exudates at 50% and 100% phosphate showed greater visual similarity in the PCA than the exudates at 25% evincing an initial sensing from the plant in response to its phosphorus demand. At the bolting stage, the three levels of phosphate fertilization had distinct root exudation patterns. These results may suggest that as the plant ages the demand for phosphorus increases, as evidenced by the differential root exudation profiles [46]. This result is in accordance with Tawaraya et al. [47], who showed that phosphorus content increased in shoot and root-dried soybean tissues as the plant developed, and that root exudate content, collected on day 1, 5, 10 and 15 of growth, increased for phosphate -depleted compared to phosphate -sufficient treatments.

Our results suggest that plants in the vegetative stage sense only the 25% phosphate treatment as being low, whereas plants at bolting stage sense both 25% and 50% phosphate as low.

This pattern of incremental phosphorus requirement during progression in plant development is common in annual plants such as *A. thaliana* [39]. For instance, *Brassica napus* L. requires an incremental supply of phosphorus at flowering onset, which is critical for protein and oil synthesis, and the development of seeds [48,49]. It is worth noting that the intermediate phosphate rate (50%) at bolting stage was largely separated (in the PCA plot) compared to 25% and 100% treatments, indicating a higher dissimilarity in root exudation composition. This could be due to the fact that the functions of plant metabolites are diverse and are not restricted only to nutrient acquisition. For instance, root exudates can be substrates, chemotactic or signaling molecules that regulate plant root and microbial interactions [50]. Such plant modulation can be specific to developmental stages [51].

The PCA data allowed for the visual determination of changes in root exudation composition between plant developmental stages and phosphate fertilization levels. We then developed a list of compounds based on those differences observed in the PCA in response to phosphorus nutrient status; and four of those compounds (i.e. 3-hydroxypropionic acid, nicotinic acid, glutamic acid and malic acid) were confirmed as phosphate solubilizers. Three (3-hydroxypropionic acid, nicotinic acid, and malic acid) out of the four compounds were significantly more abundant at the lowest phosphate rate and reduced in concentration as the amendments was elevated.

Our findings agree with a variety of studies evaluating the exudation of organic acids (e.g. malic acid) in various plant species under various phosphate availabilities [52,13]. Malic acid is a primary compound released by roots under phosphorus deficiency, but often not the most effective [52]. In contrast, nicotinic acid and glutamic acid are less abundant than malic acid or oxalic acid in plant species [46,53]. 3-Hydroxypropionic acid has not been previously associated with phosphate solubilization, however it has been described as a natural product of a plant endophytic fungus [54,55]. Further, the knowledge of the secretion of these compounds throughout plant phenology is scarce mainly because these studies are often performed in hydroponic systems limiting the root exudate collection to early stages of plant development for a variety of plant crops [52,56]. In our study, we observed that the cumulative secretion of malic acid significantly increased during low phosphate availability, but it was limited at the bolting stage. 3-Hydroypropionic acid and nicotinic acid followed the same pattern, however they were significantly secreted above control level only in the vegetative stage. Conversely, glutamic acid and nicotinic acid increased in abundance at bolting stage when phosphate levels increased. Interestingly, nicotinic acid changed cumulative secretion depending on the developmental stage. It has been reported that nicotinic acid induced flowering in *Lemna* plants [57], and that nicotinic acid can alleviate abiotic plant stresses by increasing hormone levels such as indole-3-acetic acid and gibberellic acid [58].

Based on these observations we hypothesize that plant developmental stage modulates root exudation to deal with phosphorus deficiency by three potential mechanisms: (1) Plants secrete synergistic phosphate-solubilizing compounds in stages of high phosphorus demand. In this study nicotinic acid, a moderate phosphate -solubilizer, was released in combination with 3-hydroxypropionic acid, a stronger phosphorus-solubilizer. This agrees with a recent study showing synergistic association of citrate and phytase to improve acquisition of plant unavailable phosphorus in tobacco in the vegetative stage [59]. However, this study was performed under soil conditions and not using liquid NBRIP media. (2) Plants prevent the degradation of phosphate-solubilizing compounds such as malic acid, rapidly degraded by soil microbes [60], by releasing a different compound such as 3-hydroxypropionic acid preceding a growth stage of high phosphorus demand. It has been shown that certain plants, such as lupin, can release compounds that inhibit microbial activity to reduce organic acid degradation prior to the release of organic acids. [61]. Lastly, (3) plants secrete specific compounds to mediate either

direct nutrient solubilization or the proliferation of distinct microbial taxa (with phosphate solubilizing activity) at specific growth stages (e.g. vegetative, bolting). In support of this hypothesis, root exudates can promote the activity of symbiotic microbes, such as phosphate solubilizing bacteria and siderophore-releasing bacteria and exert mobilization of non-available plant nutrients in soils at a single growth stage [62–64].

It has been estimated that organic acids constitute 5 to 10% of the total organic carbon in the soil solution. The concentration of organic anions measured in the soil solution usually ranges from 100 nM to more than 580 μM in the rhizosphere of cluster roots [65]. However, millimolar concentrations of organic anions are likely required in the soil solution to effectively increase soluble phosphate concentration especially in calcareous soils [66, 67]. Strom et al. [66] tested three organic acids (citrate, malic and oxalate) and a wide range of concentrations (1 mM to 100 mM) to evaluate its effects on the mobilization of phosphorus in calcareous soil. The results showed that the phosphorus mobilization of the tested compounds had a low efficiency and its effect varied depending on the type of organic acid, compound concentration, and pH. Further, due to the low phosphorus mobilization efficiency of those compounds it is still argued if the benefit of releasing large amounts of organic acids into the soil will exceed the cost of carbon lost by the plant, which can be seen as an unnecessary trade-off [66]. However, low efficiency organic acids can be particularly important in phosphate mobilization for calcareous soils with a limited phosphorus availability for plants. Finally, our evidence supports the above-mentioned hypothesis, that plants release a combination of compounds with different phosphorus-solubilizing efficiencies, at specific stages of growth, to deal with particular phosphorus needs.

Lastly, root exudates from liquid culture systems allow the determination of exudation rates unaltered by the soil matrix or microbial decomposition if performed under sterile conditions as we did in this study [56]. However, the quality and quantity of the root exudate profile may be impacted by the nutrient solution culture method (also known as hydroponic methods) [68]. Soil-hydroponic hybrids methods for root exudation collection are not exempt of potential physical/physiological perturbances. Thus, sterile nutrient solution culture methods remain especially important to assess temporal dynamics of root exudates.

In summary, the significance of these findings relies on the potential of customizing specific metabolites to be utilized as soil amendments under the most limiting phosphorus conditions and most demanding stage of plant growth. The role of secondary metabolites in phytoremediation efforts has been previously investigated [69] as well as the use of customized synthetic bacteria communities to modify plant phosphate accumulation [70,71]. However, the use of customized metabolites for phosphorus acquisition remains unexplored.

## Conclusions

The data collected indicate that root exudate patterns change as a response to the supply levels of phosphorus, and this change was accentuated as the plant reached maturity, when phosphate demands are higher. 3-Hydroxypropionic acid and nicotinic acid accumulated significantly at the vegetative stage under lower phosphate supplementation and was found to solubilize phosphate under both solid and liquid medium. This study sheds light on the influence of plant nutrient status and plant phenological growth stages driving the composition of plant root exudates. Future research should focus on understanding the effects of metabolites at a particular developmental stage of growth under phosphorus depleted soil, as well as to test the potential of these phosphate-solubilizing compounds in making phosphorus available for plants grown in soils saturated in unavailable phosphorus forms.

## Supporting information

**S1 Fig. Root exudate compounds diverge in response to plant developmental stage and phosphate fertilization rate. (A)** 456 compounds detected using GC-MS are plotted on the graph. PCA show dissimilarity among group of metabolites in the seedling stage at different fertilization levels: 25% (light green), 50% (light blue), 100% (green); vegetative stage: 25% (purple), 50% (pink), 100% (blue); and bolting stage: 25% (brown), 50% (olive), 100% (orange). **(B)** Data reduced to 201 annotated compounds with proper identification. PCA of compounds grouped by phosphate treatments in the seedling stage: 25% (light green), 50% (light blue), 100% (green); vegetative stage: 25% (purple), 50% (pink), 100% (blue); bolting stage: 25% (brown), 50% (olive), 100% (orange). The dotted circle indicates a cohesive group at a given fertilization level.
(DOCX)

**S2 Fig. Root exudate compounds grouped by repetitions of fertilizer level.** Treatments within plant developmental stages differ from one another, particularly the vegetative and bolting growth stages. Ellipses circle three repetitions of same fertilizer level. Color code correspond to seedling: 25% (light green), 50% (light blue), 100% (green); vegetative: 25% (purple), 50% (pink), 100% (blue); bolting: 25% (brown), 50% (olive), 100% (orange).
(DOCX)

**S3 Fig. Top 10 compounds showing changes in cumulative secretion levels in the vegetative developmental stage ($p<0.05$) in response to increasing phosphate addition (0.312, 0.625 and 1.25 mM).** Selected compounds based on PCA from vegetative 25% phosphate **(A)**, and vegetative 50% and 100% phosphate **(B)**.
(DOCX)

**S4 Fig. Top 15 compounds showing changes in cumulative secretion levels in the bolting developmental stage ($p<0.05$) in response to increasing phosphate addition (0.312, 0.625 and 1.25 mM).** Selected compounds from bolting 25% phosphate **(A)**, bolting 50% P **(B)** and bolting 100% **(C)**.
(DOCX)

## Acknowledgments

We thank Dr. James Self, manager of the Soil, Water and Plant testing laboratory at Colorado State University for their invaluable advice regarding phosphorous chemical analysis.

## Author Contributions

**Conceptualization:** Hugo A. Pantigoso, Jun Yuan, Jorge M. Vivanco.

**Formal analysis:** Hugo A. Pantigoso, Jun Yuan.

**Funding acquisition:** Jorge M. Vivanco.

**Investigation:** Hugo A. Pantigoso, Yanhui He, Qinggang Guo.

**Methodology:** Yanhui He, Qinggang Guo, Charlie Vollmer.

**Supervision:** Jorge M. Vivanco.

**Visualization:** Hugo A. Pantigoso.

**Writing – original draft:** Hugo A. Pantigoso.

**Writing – review & editing:** Hugo A. Pantigoso, Yanhui He, Jorge M. Vivanco.

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
