## [Decision Letter · Decision Letter 0]

13 Mar 2020

PONE-D-20-04662

Phosphorus assimilation effects by root exuded compounds across plant developmental stages

PLOS ONE

Dear Dr. Vivanco,

Thank you for submitting your manuscript to PLOS ONE. After careful consideration, we feel that it has merit but does not fully meet PLOS ONE’s publication criteria as it currently stands. Therefore, we invite you to submit a revised version of the manuscript that addresses the points raised during the review process.

Generally both reviewers liked your paper and neither suggested or required new experiments. However both of them made comments about the presentation and language. Reviewer 1 brings up some fairly substantial conceptual points. Please consider these and revise your text accordingly . Reviewer 2 has a large number of small but interesting comments and suggestions. The more of these that you can attend to, the better your paper will be. 

In addition, I read parts of your paper and have a few comments of my own. 

Line 1. Title. Your title is difficult to parse. Please rewrite. For example: “On the role of root exudates on the assimilation of phosphorus in young and old roots”. 

Line 26, and elsewhere. Please spell out phosphorus everywhere in the text. Do not use “P”. I am aware that many authors use this abbreviation. However repeated use is a poor argument for validity. In fact, abbreviations and acronyms make a text difficult to read. Abbreviations rarely occur in newspapers or fiction. Any time a reader encounters a symbol (P is a symbol for phosphorus), they must translate that symbol into words. Translation takes mental energy away from comprehension. Translation slows down the reader and gives nothing in return. It is one thing to have to write about a chemical with a name that is 38 syllables long. In that case, the name is just as difficult to read as the acronym. However phosphorus is a good word of the English language. In fact, phosphorus literally means ‘carrier of light’. Rather beautiful. Spell it out. 

Line 26. “… have not been conducted…” This wording implies that you have read every paper ever published. Use less extreme wording, such as “…rarely if ever…”

Line 29. You are using the word ‘metabolome’, to refer to root exudates. This is misleading and confusing. The word metabolome refers to all of the metabolites in the cell or organism. However here, you are not measuring the metabolome. Instead you are measuring exudates. These compounds number about a dozen, far less than the threshold for ‘omics’.  Please remove the word ‘metabolome’ throughout the paper and instead talk about ‘total exudates’ or the equivalent.  For example, the sentence staring at line 29 can read: “The composition of root exudates changed in response…”

Lines 29 and 34. You write ‘in vitro’ conditions. What does this mean? Instead say what the conditions are. By the way, typically in vitro implies isolated components, cells or cell fragments. I have never seen in vitro used for whole organisms. Also you write ‘solid’ and ‘liquid media conditions’. Again, be specific, say what the solid and liquid media are. 

Line 31. What does ‘respond negatively’ mean? If you mean decreased in quantity, just say so. 

Line 66. Based on long-established rules for scientific nomenclature, the word “*Arabidopsis*” (Capital A, italic font) means the genus. No matter how many illiterate molecular biologists make this mistake, rules of taxonomic nomenclature remain in force until the international committee decrees otherwise. Longstanding practice allows “arabidopsis” (lower case a, Roman font) as the common name of our friendly lab weed; however, many journals will auto-correct this by adding a capital and italics. I am not sure about PLoS One. Thus, either use ‘arabidopsis’ and hope the journal lets it stand, or use *A. thaliana*(italics) (but *Arabidopsis thaliana*at first mention). 

Line 83. This should read “*Arabidopsis thaliana*L. (Heynh) wild-type Columbia seed…” It is customary in the Material and Methods to give the full Latin binomial along with the taxonomic authority. And unless you sequenced your Columbia line, you should drop the “0”.

Line 84, and elsewhere. Do not use circle-R, or TM, or other commercial symbols. Those marks exist to protect consumers (not manufacturers) from fraud. Thus if you sell a product that you say contains X circle-R, you are defrauding the consumer if you substitute something cheaper for X. There is no such issue here. In fact you are not selling anyone anything (you are paying). 

Line 98. I don’t understand the meaning of the word ‘power’ here. 

You embedded the figure legends into the results text but not the figures. I think this is awkward. Please put the figure legends with the figures, preferably at the end of the text. Also for the principal component analyses, please simply call your x and y axes “PC1” and “PC2” and put the percentages of explained variance in the figure legend. 

In figure 3A, the names of some compounds are capitalized but others are not. And “control” is in all caps. This is bizarre. I suggest that none of them should be capitalized, but whatever you do please do the same for all. And note that if you choose to capitalize the first letter, this should be done even for those compounds that start with a number (e.g., “5-Aminovaleric acid”). Also, please put the name of the horizontal axis below, next to the numbers (or put the numbers above, next to the name; the point is, the numbers and the name belong together. 

Also please remember that the abbreviation for liter is L (not l). And to always put a space between the number and the unit (except when the unit is the degree sign or the percent sign). 

In Figure 4, there are problems with the y-axes. Many of them have huge numbers. But no units. I think peak area is arbitrary so you could use 1, 2, 3. If it is needed to compare the quantity between states, then you could have the smallest numbers be 1, 2, 3 and the other states relative to that. Also many of the graphs do not start at zero. But some of them do. This is misleading. They should all start at zero. The names of the axes could be in a larger font. 

We would appreciate receiving your revised manuscript by Apr 27 2020 11:59PM. To enhance the reproducibility of your results, we recommend that if applicable you deposit your laboratory protocols in protocols.io, where a protocol can be assigned its own identifier (DOI) such that it can be cited independently in the future. For instructions see: http://journals.plos.org/plosone/s/submission-guidelines#loc-laboratory-protocols

We look forward to receiving your revised manuscript.

Kind regards,

Tobias Isaac Baskin

Academic Editor

PLOS ONE

Journal Requirements:

Reviewers' comments:

Reviewer's Responses to Questions

**Comments to the Author**

1. Is the manuscript technically sound, and do the data support the conclusions?

Reviewer #1: Yes

Reviewer #2: Partly

2. Has the statistical analysis been performed appropriately and rigorously? 

Reviewer #1: Yes

Reviewer #2: Yes

3. Have the authors made all data underlying the findings in their manuscript fully available?

Reviewer #1: Yes

Reviewer #2: Yes

4. Is the manuscript presented in an intelligible fashion and written in standard English?

Reviewer #1: Yes

Reviewer #2: Yes

5. Review Comments to the Author

Reviewer #1: Two minor and three major critics

minor:

1. Root exudates depending on plant develpment were reported by Keerthisinghe et al. 1998, Plant Cell Environ., 21, 467- 478 and by Neumann et al., 1999, Planta, 208, 373- 382.

2. L 278 ff. Misleading since P and carboxylate concetrations must be clearly separated.

major:

1. Misleading: P starvation response 50 µM P. Cultivation in non buffered systems, agar or solution means that a solution threshald is only valid for a special volume. In soil P is buffered and P starvation values are below 2 µM P , see Föhse te al., 1988, Plant Soil, 110, 101- 109.

2. Decisive for the solubilizing ability of root exudates is the relation between soil solid P and the quatity of exudates (see in detail Gerke, 2015, cited in the ms). Solution concentrations are not an appropriate measure of exudate efficiency. Mesurement of pH changes as a central factor of P mobilization from Ca-P forms?

3. The relevance of the results with respect to P solubilization is questionable since the P form is rather unrepresentative and the exudate quantity is rather high.

Reviewer #2: General comments:

The manuscript “Phosphorus assimilation effects by root exuded compounds across plant developmental stages” characterises root exudates across growth stages and P-levels for Arabidopsis. Interesting contrasts were found. The authors select some important exuded compounds and go on to test their ability at solubilising P in soil-analogues individually and together. This was a well written manuscript, with well thought out experiments, executed well leading to interesting results. I really enjoyed reading this manuscript, I learnt a lot that is relevant to my work, and appreciated the work that went into it. I have some general comments that I think would improve the presentation and some of the discussion.

1) I think in the materials and methods the presentation of the PCA is sometimes over complicated, and could be made simpler. My detailed comments below would address this.

2) The presentation of the PCA results could be improved. I think showing all the data with no groupings used on the PC-axis (with different colours and marker styles and legends!) would make it much clearer an. This could come at the expense of some of the plots where treatments are grouped. Also make it clear if a PCA is done per grouping or not. Again, many of my detailed comments try and address this.

3) The discussion would be improved by quickly pointing out the agricultural/ecological significance (or lack of) of the results. For example 7mM of acids order of magnitude more than what is found in the rhizosphere. See the Eva Oburger papers I referenced. Hence the solubilisation seen here might not appear in the field. One of our group's papers found that P solubilisation by a single root exuding citrate at a realistic rate actually made no difference to P uptake by the root https://doi.org/10.1007/s11104-019-04376-4 (dont feel you have to reference this). It also might be worth pointing out how similar NBRIP media is to soil, solubilisation in this media might not be the same as soil. Also see my detailed comment regarding your proposed mechanism number 1.

Sorry, I have a lot of detailed comments, however many of them are pointing out the same thing and many are complimentary. I hope they aren’t too hard to deal with.

Detailed comments:

L25: roots->root

Abstract is clear and well written.

The paragraph starting line 67 doesn’t segway smoothly into that starting on 73 because root exudates have been shown to be less important for good P conditions (not to say that the paragraph on L67 isnt useful introductory information). Calling back to the ‘legacy P’ argument here would make it smoother.

L90: ‘gradient’ isn’t the best word here. It makes it seem like the growth media each plant is grown in has a gradient of P conditions in it. Is ‘The low P conditions used in this study….’ Better?

L129: ‘…which compounds had the highest correlation with these components through the magnitude of the variance’. ‘correlation’ is unclear here. Do you mean which two original basis vectors (compounds) explained the most variance? This would be an easier way of explaining it.

L133: ‘The largest…’ Is it not that these compounds explain the most variance? Its best to write these in terms of what percent of the original variance they explain (before dimension reduction).

L134: differential isn’t the right word here.

I think the end of this paragraph can be explained in a simpler way by saying: “We determined which compounds explained the most variance for each P level and growth stage. The contrasts in these ‘most principle compounds’ over growth stage and P level indicated differences in exudation depending on these factors” Or something like this.

L144: This needs more explanation this is quite a large range, how did you ‘adjust’ the concentrations? The reader needs to know how this is done to interpret the results. Did you take concentrations from these references to calibrate your values? Due to the variability in reported exudate concentration due to sampling technique (https://doi.org/10.1016/j.rhisph.2018.06.004) ‘exudate concentration’ is relative to the study. Probably needs a mention in the discussion.

Side note: Could be worth mentioning that you didn’t adjust the concentrations for the PCA, when I first read it I thought you had.

L152: What concentrations did you add to test the P solubilising ability of the compounds? Were they comparable to rhizospshere concentrations? Knowing this puts this into an ecological context for the reader. Is the visual inspection for solubility standard? Add a reference if so. Otherwise explain the reasoning.

Table 1: were they diluted to 100mM before PCA? Would this effect PCA? (see my side note above). 100mM is a really high concentration for most root exudates in the rhizosphere no? I know for most organic acids and amino acids I would expect micro molar concentrations in the rhizosphere, see https://doi.org/10.1016/j.envexpbot.2012.11.007

L171: again this a high concentration for the rhizosphere, I would expect a discussion on the ecological relevance later.

L181: I think ‘cumulative’ effect is better than ‘additive’. Additive makes it seem like you could add up the effects individually to determine to summed effect.

L200: unclear what ‘distribution’ means here. After looking at FigS1 we see it’s the groupings after plotting on the first two principle components. This needs to be made clearer. I think the author means the separation of growth stage and P treatment by PCA persists after throwing away the non-annotated compounds. Which implies the non-annotated compounds explained little of the variance, which I believe justifies the author’s conclusion. Were the principle components determined twice i.e. before and after throwing away the non-annotated compounds? FigS1 could use a legend saying what the other colours are.

Fig 1A also needs a legend.

In Fig 1A did you group by developmental stage, do PCA then plot? If this is the case, the % variance explained by the PCs reported in line 203 isnt correct.

Same goes for Fig 1B but by P level? The different percent of variance explained by components suggests this is the case. Need legend for Fig 1B also.

I would use colours to indicate developmental stage and stars/dots/crosses to indicate P-levels (or vice versa). This and a legend would make fig 1 a lot easier to understand.

Fig 1A caption is wrong “201 annotated compounds with proper identification detected using GC-MS are plotted on the Principal Component Analysis (PCA) graph”. The compounds aren’t plotted. The compounds are the 201-axis, the dimension is then reduced to 2-axis with PCA and the locations of the growth stages are plotted.

L205: “We determined that the plant’s developmental stage was responsible for

the separation of the compounds in three marked groups” to convince the reader that its not P level you would need to not group by anything and plot all the data with stars/dots/crosses and colors.

All these figures desperately need legends (Fig1, S1, S2) and clearly state if there is a PCA per grouping.

L230: explained the most variance.

L231: ‘correlation’ isn’t really defined here. Its more that the compounds which point in the most similar direction to the two principle components, I would get rid of the bit after the comment.

Change this sentence to: For each of these treatments, we found the five top compounds that explained the largest proportion of the variance.

Figs S3-S4 are really intresting! Great results

Same goes for Fig 2 and 3A

L254: does the 1.75 7 and 28 mM mean that the sum of the concentrations or each have that concentration? Fig 3 caption makes this clear but also make it clear in the manuscript.

L255: The 1.75M is really interesting, 7mM of them mixed does worse than 7mM of any individually

Again, really interesting results with Fig 4, very hard to interpret though there is lots going on.

L300: …’where optimum conditions are scare’ -> where nutrients are scarce ??

L304: These papers might be useful for this discussion on maize need for P early on with respect to yield: https://doi.org/10.1007/s11104-011-0814-y. https://www.nrcresearchpress.com/doi/pdfplus/10.4141/P00-093) Take them or leave them

L309: ‘similarity’ do you mean they grouped close together in the PSA? To talk about similarity you would need to use a norm between the treatments.

L323: I think your mention of other functions of exudates is a good explanation and something other authors overlook.

346: that ‘the’ shouldn’t be there

L351: new paragraph

L352: mechanism 1. Your results suggest there is no synergy (in fact the opposite) in terms of P solubilisation by root exudates, see my comment about L255, at least for the 4 acids you picked out. But this doesn’t disprove your mechanism: I think I read somewhere that certain exudates solubilise P from certain soil surfaces, Al oxides, Fe oxides etc (possibly a paper by Gerke about citrate and malate?) Maybe this is why they exude a mixture of acids? Did the soil-analogue you used have this variability in mineral surfaces which soil has?

Nonetheless, I think this deserves some more discussion.

L357. I like point 2, I had never heard of this before.

I don’t think Table S1 is mentioned in the manuscript

6. PLOS authors have the option to publish the peer review history of their article (what does this mean?). If published, this will include your full peer review and any attached files.

Reviewer #1: Yes: Dr. Jörg Gerke

Reviewer #2: No

---

## [Author Response · Author response to Decision Letter 0]

6 Apr 2020

Reviewers responses:

Line 1. Title. Your title is difficult to parse. Please rewrite. For example: “On the role of root exudates on the assimilation of phosphorus in young and old roots”. 

RESPONSE: Thank you. As suggested, the title was rewritten and now it reads as follow: ‘Role of root exudates on assimilation of phosphorus in young and old Arabidopsis thaliana plants

 Line 26, and elsewhere. Please spell out phosphorus everywhere in the text. Do not use “P”. I am aware that many authors use this abbreviation. However repeated use is a poor argument for validity. In fact, abbreviations and acronyms make a text difficult to read. Abbreviations rarely occur in newspapers or fiction. Any time a reader encounters a symbol (P is a symbol for phosphorus), they must translate that symbol into words. Translation takes mental energy away from comprehension. Translation slows down the reader and gives nothing in return. It is one thing to have to write about a chemical with a name that is 38 syllables long. In that case, the name is just as difficult to read as the acronym. However phosphorus is a good word of the English language. In fact, phosphorus literally means ‘carrier of light’. Rather beautiful. Spell it out. 

RESPONSE: Thanks for the suggestion and wonderful comments particularly related to “carrier of light”. The abbreviation for phosphorus was changed for the full word throughout the manuscript. 

Line 26. “… have not been conducted…” This wording implies that you have read every paper ever published. Use less extreme wording, such as “…rarely if ever…”

RESPONSE: Thanks for the suggestion. We have changed ‘have not been conducted’ for ‘remain scarce’. 

Line 29. You are using the word ‘metabolome’, to refer to root exudates. This is misleading and confusing. The word metabolome refers to all of the metabolites in the cell or organism. However here, you are not measuring the metabolome. Instead you are measuring exudates. These compounds number about a dozen, far less than the threshold for ‘omics’. Please remove the word ‘metabolome’ throughout the paper and instead talk about ‘total exudates’ or the equivalent. For example, the sentence staring at line 29 can read: “The composition of root exudates changed in response…”

 RESPONSE: Thanks. We agreed with the editor. We replaced the word metabolome for total root exudates and/or total exudate profile throughout the manuscript.

Lines 29 and 34. You write ‘in vitro’ conditions. What does this mean? Instead say what the conditions are. By the way, typically in vitro implies isolated components, cells or cell fragments. I have never seen in vitro used for whole organisms. Also you write ‘solid’ and ‘liquid media conditions’. Again, be specific, say what the solid and liquid media are. 

RESPONSE: 

Thank you for the recommendation. To make the distinction, we have replaced the word in vitro for ‘sterile nutrient solution’.

The type of media described as solid and liquid is a microbiological growth medium for screening phosphate solubilizing microorganisms known as NBRIP (National Botanical Research Institute’s phosphate) growth media. This information appears in the material and methods. In line 29-34 (now 36) we have specified the type of liquid and solid media and it reads now as: “solid and liquid NBRIP media conditions”.

Line 31. What does ‘respond negatively’ mean? If you mean decreased in quantity, just say so. 

RESPONSE: Thank you. Changes were made accordingly and now the sentence reads as follows: ‘It was found that specific metabolites decreased in quantity related to phosphorus supplementation at specific stages of development.’

Line 66. Based on long-established rules for scientific nomenclature, the word “Arabidopsis” (Capital A, italic font) means the genus. No matter how many illiterate molecular biologists make this mistake, rules of taxonomic nomenclature remain in force until the international committee decrees otherwise. Longstanding practice allows “arabidopsis” (lower case a, Roman font) as the common name of our friendly lab weed; however, many journals will auto-correct this by adding a capital and italics. I am not sure about PLoS One. Thus, either use ‘arabidopsis’ and hope the journal lets it stand, or use A. thaliana(italics) (but Arabidopsis thalianaat first mention). 

RESPONSE: Thanks for the observation. Arabidopsis thaliana was first mentioned in the abstract and ‘Arabidopsis’ word was replaced by A. thaliana throughout the manuscript. 

Line 83. This should read “Arabidopsis thalianaL. (Heynh) wild-type Columbia seed…” It is customary in the Material and Methods to give the full Latin binomial along with the taxonomic authority. And unless you sequenced your Columbia line, you should drop the “0”.

RESPONSE: Thanks for the observation. The sentence was modified and now it reads as follows: ‘A. thaliana (L.) Heynh wild-type Columbia seeds’.

Line 84, and elsewhere. Do not use circle-R, or TM, or other commercial symbols. Those marks exist to protect consumers (not manufacturers) from fraud. Thus if you sell a product that you say contains X circle-R, you are defrauding the consumer if you substitute something cheaper for X. There is no such issue here. In fact you are not selling anyone anything (you are paying). 

RESPONSE: Thank you. Changes were made accordingly.

Line 98. I don’t understand the meaning of the word ‘power’ here. 

RESPONSE: Thank you. The word power is not necessary. It was deleted.

You embedded the figure legends into the results text but not the figures. I think this is awkward. Please put the figure legends with the figures, preferably at the end of the text. Also for the principal component analyses, please simply call your x and y axes “PC1” and “PC2” and put the percentages of explained variance in the figure legend. 

 RESPONSE: As suggested, figures and legends are together in a separate file uploaded right after the text. 

All principal component analyses have been modified and the explained variance is now shown in the legend. 

In figure 3A, the names of some compounds are capitalized but others are not. And “control” is in all caps. This is bizarre. I suggest that none of them should be capitalized, but whatever you do please do the same for all. And note that if you choose to capitalize the first letter, this should be done even for those compounds that start with a number (e.g., “5-Aminovaleric acid”). Also, please put the name of the horizontal axis below, next to the numbers (or put the numbers above, next to the name; the point is, the numbers and the name belong together. 

Also please remember that the abbreviation for liter is L (not l). And to always put a space between the number and the unit (except when the unit is the degree sign or the percent sign). 

RESPONSE: Thanks for the observation. The first letters of all compounds have been capitalized, and the horizontal axis is now next to the numbers. The abbreviation L and the space between the number and the unit was also modified as suggested. 

 In Figure 4, there are problems with the y-axes. Many of them have huge numbers. But no units. I think peak area is arbitrary so you could use 1, 2, 3. If it is needed to compare the quantity between states, then you could have the smallest numbers be 1, 2, 3 and the other states relative to that. Also many of the graphs do not start at zero. But some of them do. This is misleading. They should all start at zero. The names of the axes could be in a larger font. 

RESPONSE: Thank you for the observation. The y-axis values show the magnitude of the cumulative secretion level of the compound per each stage. Those values are different based on the compounds. Thus, standardizing the values for all compounds will make some compounds hard to asses due to low levels. This set of plots aimed to provide to the reader with a visual aid to see significant differences by compound between the phosphorous treatments. 

We think keeping the peak area unit as it is will help the reader to determine the degree of change (of the compound) in response to P amendment as well as to locate the compound that was the most affected by phosphorus amendment. 

With respect to graphs not starting at zero: the compounds with statistical significance are properly labeled which will help the reader to see which of those secretion levels are highly responsive to phosphorus addition. We do not think this is misleading and we would rather keep it as it is. 

Reviewer #1: Two minor and three major critics

minor:

1. Root exudates depending on plant develpment were reported by Keerthisinghe et al. 1998, Plant Cell Environ., 21, 467- 478 and by Neumann et al., 1999, Planta, 208, 373- 382.

RESPONSE: We appreciate your valuable comments. Both papers are very relevant for our work, and we have included references about that work in our manuscript. We modified the abstract to state that: ‘However, studies addressing the variability of roots exudates involved in phosphorus solubilization across plant developmental stages remain scarce’. Instead of saying that these studies ‘have not been conducted’. Having said this, in contrast with Keerthisinghe et al. (1998) and Neumann et al., (1999), our study showed effects of P supply on non-proteoid plant-roots such as Arabidopsis. 

Neumann et al., (1999) support our findings showing that phosphorus mobilizing compounds such as citric acid was predominantly restricted to mature root clusters in the later stages of P deficiency. However, they conclude that this mechanism is ‘predominantly confined to proteoid-roots’ which is not the case for Arabidopsis as shown in our study. We have incorporated this to the discussion section.

2. L 278 ff. Misleading since P and carboxylate concetrations must be clearly separated.

RESPONSE: Thank you for the observation. We have rewritten this sentence to better describe how cumulative secretion of different carboxylates vary depending on the treatment (phosphorus level) and to make the distinction between phosphorus and carboxylates. Now the sentence reads as follows: ‘Cumulative secretion of malic acid was reduced significantly from 0.3125 to 0.625 mM phosphorus treatments, and then incremented its cumulative secretion for the highest phosphorus treatment (1.25 mM). However, secretion levels of malic acid for the two highest phosphorus treatments (0.625 and 1.25 mM) were below the value for the lowest phosphorus rate (0.3125 mM) (Fig 4A).’

major:

1. Misleading: P starvation response 50 µM P. Cultivation in non buffered systems, agar or solution means that a solution threshald is only valid for a special volume. In soil P is buffered and P starvation values are below 2 µM P , see Föhse te al., 1988, Plant Soil, 110, 101- 109.

RESPONSE: Thank you for the clarification. As per your suggestion, we clarified that plant starvation response to phosphorus is activated at values below 2 µM P. We have added Föhse et al., (1988) reference to the manuscript and the sentence now reads as follows: ‘The three phosphorus levels used in this study did not induce the plant starvation response, which is generally activated at values below 2 μM of phosphorus in the soil solution [23].

2. Decisive for the solubilizing ability of root exudates is the relation between soil solid P and the quatity of exudates (see in detail Gerke, 2015, cited in the ms). Solution concentrations are not an appropriate measure of exudate efficiency. Mesurement of pH changes as a central factor of P mobilization from Ca-P forms?

RESPONSE: We appreciate your valuable comment. We are aware that a combination of factors including carboxylate efflux, the accumulation of carboxylate in the rhizosphere and the chemistry of phosphorus mobilization in the soil phase affect the solubilization ability of root exudates (Gerke, 2015). We acknowledge that we did not take into account these variables, because our study was entirely done in liquid culture to test solubilization ability of the selected compounds, not necessarily efficiency which would require the measurement of abovementioned factors. However, we have added a paragraph in the discussion section that address pros and cons of root exudate studies in liquid conditions (our study). 

 The paragraph added reads as follows: ‘Lastly, root exudates from liquid culture systems allow the determination of exudation rates unaltered by the soil matrix or microbial decomposition if performed under sterile conditions as we did in this study (Oburger and Jones, 2015). However, the quality and quantity of the root exudate profile may be impacted by the nutrient solution culture method (also known as hydroponic methods) (Oburger et al., 2013). Soil-hydroponic hybrids methods for root exudation collection are not exempt of potential physical/physiological perturbances. Thus, sterile nutrient solution culture methods remain especially important to assess temporal dynamics of root exudates.’

3. The relevance of the results with respect to P solubilization is questionable since the P form is rather unrepresentative and the exudate quantity is rather high.

RESPONSE: Thank you for your observation. It would be helpful if the reviewer elaborates on his comment. Why are the P form is unrepresentative? Is it because calcium phosphate is not necessarily predominant in all agricultural soils? Calcareous soils constitute over one-third of the world’s land area. Why you consider the root exudate quantity as high, relative to what? Root exudates concentration varies a lot across P levels, developmental stage and on a per compounds basis in our study.

Reviewer #2: General comments:

The manuscript “Phosphorus assimilation effects by root exuded compounds across plant developmental stages” characterises root exudates across growth stages and P-levels for Arabidopsis. Interesting contrasts were found. The authors select some important exuded compounds and go on to test their ability at solubilising P in soil-analogues individually and together. This was a well written manuscript, with well thought out experiments, executed well leading to interesting results. I really enjoyed reading this manuscript, I learnt a lot that is relevant to my work, and appreciated the work that went into it. I have some general comments that I think would improve the presentation and some of the discussion.

1) I think in the materials and methods the presentation of the PCA is sometimes over complicated, and could be made simpler. My detailed comments below would address this.

RESPONSE: Thanks for the valuable observations. Detail corrections regarding the PCA plots have been addressed in each of your comments below. 

2) The presentation of the PCA results could be improved. I think showing all the data with no groupings used on the PC-axis (with different colours and marker styles and legends!) would make it much clearer an. This could come at the expense of some of the plots where treatments are grouped. Also make it clear if a PCA is done per grouping or not. Again, many of my detailed comments try and address this.

RESPONSE: Thanks for the valuable observations. Detail corrections regarding the PCA plots have been made and addressed in each of your comments below. 

3) The discussion would be improved by quickly pointing out the agricultural/ecological significance (or lack of) of the results. For example 7mM of acids order of magnitude more than what is found in the rhizosphere. See the Eva Oburger papers I referenced. Hence the solubilisation seen here might not appear in the field. One of our group's papers found that P solubilisation by a single root exuding citrate at a realistic rate actually made no difference to P uptake by the root https://doi.org/10.1007/s11104-019-04376-4 (dont feel you have to reference this). It also might be worth pointing out how similar NBRIP media is to soil, solubilisation in this media might not be the same as soil. Also see my detailed comment regarding your proposed mechanism number 1.

RESPONSE: Thank you for the observations. We have added a paragraph that discusses the relevance and ecological significance of organic acids as phosphorus mobilizers and the relevance of the root exudate analysis in liquid cultures. 

The new paragraph reads as follows: ‘It has been estimated that organic acids constitute 5 to 10 % of the total organic carbon in the soil solution. The concentration of organic anions measured in the soil solution usually range from 100 nM to more than 580 uM in the rhizosphere of cluster roots (Jones, 1998). However, millimolar concentrations of organic anions are likely required in the soil solution to effectively increase soluble P concentration especially in calcareous soils (Strom et al. 2005; Ryan and Jones, 2001). Strom et al. (2005) tested three organic acids (citrate, malic and oxalate) and a wide range of concentrations (1 mM to 100 mM) to evaluate its effects on the mobilization of phosphorus in calcareous soil. The results showed that the phosphorus mobilization of the tested compounds had a low efficiency and its effect varied depending on the type of organic acid, compound concentration, and pH. Further, due to the low phosphorus mobilization efficiency of those compounds it is still argued if the benefit of releasing large amounts of organic acids into the soil will exceed the cost of carbon lost by the plant, which can be seen as an unnecessary trade-off (Strom et al. 2005). However, low efficiency organic acids can be particularly important in phosphorus mobilization for calcareous soils with a limited phosphorus availability for plants. Finally, our evidence supports the above-mentioned hypothesis, that plants release a combination of compounds with different phosphorus-solubilizing efficiencies, at specific stages of growth, to deal with particular phosphorous needs.’

Sorry, I have a lot of detailed comments, however many of them are pointing out the same thing and many are complimentary. I hope they aren’t too hard to deal with.

Detailed comments:

L25: roots->root

RESPONSE: Thank you. Please see correction in line 28. 

Abstract is clear and well written.

The paragraph starting line 67 doesn’t segway smoothly into that starting on 73 because root exudates have been shown to be less important for good P conditions (not to say that the paragraph on L67 isnt useful introductory information). Calling back to the ‘legacy P’ argument here would make it smoother.

RESPONSE: Thank you for the observation. We considered your suggestion and have improved the transitioning sentence between the two paragraphs. Now it reads as follows: ‘Due to the variation of phosphorus demand during the plant’s lifetime, it becomes necessary to fully understand fluctuations of root exudates as a means to solubilize the phosphorous present in the P legacy of the soil.’

L90: ‘gradient’ isn’t the best word here. It makes it seem like the growth media each plant is grown in has a gradient of P conditions in it. Is ‘The low P conditions used in this study….’ Better?

RESPONSE: The authors used gradient to refer to the three concentrations of phosphorous used in the study. For clarity, we changed it to: ‘The three phosphorus levels used in this study did not stimulate the plant starvation response, which is generally activated around 2 μM of P [23].’

L129: ‘…which compounds had the highest correlation with these components through the magnitude of the variance’. ‘correlation’ is unclear here. Do you mean which two original basis vectors (compounds) explained the most variance? This would be an easier way of explaining it.

RESPONSE: Thank you for the observation. No, in this sentence (now line 140) we refer to the magnitude of the loadings of the compound (variance of the compound secretion level) not to the PCA explained variance. We elaborate on the meaning in more detail in the two following sentences in the manuscript. 

L133: ‘The largest…’ Is it not that these compounds explain the most variance? Its best to write these in terms of what percent of the original variance they explain (before dimension reduction).

RESPONSE: Here we refer to the variance of the individual compounds based on the secretion level (loadings). Compounds with the largest loadings drive the PCA variance and therefore were selected for further testing. Please see line 131 to 146 for more detail.

L134: differential isn’t the right word here.

I think the end of this paragraph can be explained in a simpler way by saying: “We determined which compounds explained the most variance for each P level and growth stage. The contrasts in these ‘most principle compounds’ over growth stage and P level indicated differences in exudation depending on these factors” Or something like this.

RESPONSE: Thank for your suggestion. We have modified the sentence for clarity and now it reads as follows: ‘this method allowed us to determine which compounds explained most of the variance across the fertilizer levels and for each of the plant’s growth stages.’ 

L144: This needs more explanation this is quite a large range, how did you ‘adjust’ the concentrations? The reader needs to know how this is done to interpret the results. Did you take concentrations from these references to calibrate your values? Due to the variability in reported exudate concentration due to sampling technique (https://doi.org/10.1016/j.rhisph.2018.06.004) ‘exudate concentration’ is relative to the study. Probably needs a mention in the discussion.

Side note: Could be worth mentioning that you didn’t adjust the concentrations for the PCA, when I first read it I thought you had.

RESPONSE: Thanks for your observations. As stated in the text, we used a range of concentrations reported in the literature as a reference to select the concentration of the compounds tested in this study. The word ‘adjust’ has been replaced by the word ‘select’ for clarity. 

As you well pointed out, the range of concentrations of this compound is wide. We cited 3 papers that support that the concentrations used in our experiments are present in the rhizosphere (Jones, 1998; Veneklaas et al., 2003 and Ström et al., 2005). In addition, we have added a paragraph in the discussion where we elaborate on the concentrations of compounds selected for this study (please see previous answer to your comment) and included the paper that you suggested: Oburger and Jones, 2018. 

L152: What concentrations did you add to test the P solubilising ability of the compounds? Were they comparable to rhizospshere concentrations? Knowing this puts this into an ecological context for the reader. Is the visual inspection for solubility standard? Add a reference if so. Otherwise explain the reasoning.

RESPONSE: Thank you for the comments. We direct the reviewer to the methodology sub-section (149-160) where the rationale for selecting the concentrations to test P solubilization and the actual concentrations are stated; as well as the paper showing the efficacy of the qualitative technique used in our study. Please see reference number 30 in the reference list (Nautiyal et al., 1999. FEMS microbiology Letters, 170(1), 265-270).

Table 1: were they diluted to 100mM before PCA? Would this effect PCA? (see my side note above). 100mM is a really high concentration for most root exudates in the rhizosphere no? I know for most organic acids and amino acids I would expect micro molar concentrations in the rhizosphere, see https://doi.org/10.1016/j.envexpbot.2012.11.007

RESPONSE: Thanks for the observation. Indeed, PCA show total root exudates as a response to phosphorus levels. We agree, 100 mM is in the higher end of the concentrations. This concentration was used only in the qualitative experiment (Petri dish). Several lower concentrations were used as well, and we decided to use 100 mM for this qualitative analysis which aimed to provide an initial visual tool to narrow down a long list of compounds that could solubilize P. Please note that we reduced this concentration to 7 mM and the compounds still showed phosphorus solubilization. 

L171: again this a high concentration for the rhizosphere, I would expect a discussion on the ecological relevance later.

RESPONSE: This question was addressed above.

L181: I think ‘cumulative’ effect is better than ‘additive’. Additive makes it seem like you could add up the effects individually to determine to summed effect.

RESPONSE: Thank you. We replaced the word additive for the word ‘cumulative’ to better describe the meaning. 

L200: unclear what ‘distribution’ means here. After looking at FigS1 we see it’s the groupings after plotting on the first two principle components. This needs to be made clearer. I think the author means the separation of growth stage and P treatment by PCA persists after throwing away the non-annotated compounds. Which implies the non-annotated compounds explained little of the variance, which I believe justifies the author’s conclusion. Were the principle components determined twice i.e. before and after throwing away the non-annotated compounds? FigS1 could use a legend saying what the other colours are.

RESPONSE: Thank you. For clarity, the word distribution has been replaced by the word “grouping”. In addition to indicating that the PCA was conducted with the data after removal of the non-annotated compounds, we have modified this sentence and now it reads as follow: ‘The variability in our data after subtracting the non-annotated compounds was analyzed using a principal component analysis (PCA) where variability of component 1 (PC1) accounted for 29.8%, while component 2 (PC2) accounted for 21.7%.’

PCA from figure 1 is identical than supplementary figure 1. Both depict the overall dissimilarity of compound groups across phosphorus levels and growth stages. Both figures show detailed information in the legend now. 

Fig 1A also needs a legend.

RESPONSE: Thank you. Figure 1A has now a legend with the suggested information. 

In Fig 1A did you group by developmental stage, do PCA then plot? If this is the case, the % variance explained by the PCs reported in line 203 isnt correct.

RESPONSE: Thank you for the comment. For figure 1 as well as for supplementary figure 1, no grouping was performed before constructing the PCA. The complete data set from three developmental stages and 3 levels were plotted and analyzed at the same time without previous modification. As suggested, the explained variance of figure 1A was added to the legend. For more detail, please see line 132-144.

Same goes for Fig 1B but by P level? The different percent of variance explained by components suggests this is the case. Need legend for Fig 1B also.

RESPONSE: In figure 1A, we present the three plant stages and three phosphorus levels (all data). Figure 1B presents data for three phosphorus levels within the vegetative stage only, and figure 1C presents the three phosphorus levels within the vegetative stage only. The explained variance for each PCA has been added to the figure legend.

I would use colours to indicate developmental stage and stars/dots/crosses to indicate P-levels (or vice versa). This and a legend would make fig 1 a lot easier to understand.

Fig 1A caption is wrong “201 annotated compounds with proper identification detected using GC-MS are plotted on the Principal Component Analysis (PCA) graph”. The compounds aren’t plotted. The compounds are the 201-axis, the dimension is then reduced to 2-axis with PCA and the locations of the growth stages are plotted.

RESPONSE: Thank you for the clarification. Your observation is technically correct. We have replaced the word plotted. Now, the sentence reads as follows: ‘201 annotated compounds with proper identification detected using GC-MS were analyzed using a Principal Component Analysis (PCA) graph.’ 

With respect to editing the colors. The phosphorus levels and growth stages are color-coded, and the information related to each treatment is explained in the legend. We would rather keep the graphs as they are currently. 

L205: “We determined that the plant’s developmental stage was responsible for

the separation of the compounds in three marked groups” to convince the reader that its not P level you would need to not group by anything and plot all the data with stars/dots/crosses and colors.

All these figures desperately need legends (Fig1, S1, S2) and clearly state if there is a PCA per grouping.

RESPONSE: Thank you. This comment was addressed in the previous answers. 

L230: explained the most variance.

RESPONSE: Thank you. Following your suggestion, we have corrected this sentence and now it reads as follows: ‘For each of these treatments, we found the five top compounds that explained the largest proportion of the variance in the principal components (Table 1).’

L231: ‘correlation’ isn’t really defined here. Its more that the compounds which point in the most similar direction to the two principle components, I would get rid of the bit after the comment.

Change this sentence to: For each of these treatments, we found the five top compounds that explained the largest proportion of the variance.

RESPONSE: Thanks. We added this suggestion in the previous response. 

Figs S3-S4 are really intresting! Great results

Same goes for Fig 2 and 3A

RESPONSE: Thank you.

L254: does the 1.75 7 and 28 mM mean that the sum of the concentrations or each have that concentration? Fig 3 caption makes this clear but also make it clear in the manuscript.

RESPONSE: Good observation. Each compound added to the pool had 1.75 mM, 7 mM and 28 mM concentration. We have added this detail in the manuscript. As explained in the methodology section line 189-195: ‘In order to determine the potential cumulative effect of 3-hydroxypropionic acid, malic acid, nicotinic acid, and glutamic acid they were combined and the available phosphorus (mg l-1) was determined by OES-ICP. A phosphorus gradient that included the previous tested concentration (7 mM per compound), a higher (28 mM per compound) and a lower (1.75 mM per compound) concentration was tested in order to compare if the combination of compounds would equal or surpass the effect of a single compound. Each compound added to the pool had 1.75 mM, 7 mM and 28 mM concentration. Thus, the combination effect of four compounds were tested in a liquid NBRIP medium.’

L255: The 1.75M is really interesting, 7mM of them mixed does worse than 7mM of any individually

Again, really interesting results with Fig 4, very hard to interpret though there is lots going on.

RESPONSE: Each compound has a different molarity; therefore we cannot say that the 4 compounds at 1.75 mM add up to 7mM. We conclude that 1) compounds in combination have an additive effect, and that 2) the addition of one does not inhibit the effect of the other. 

L300: …’where optimum conditions are scare’ -> where nutrients are scarce ??

RESPONSE: Thank you. We have corrected this sentence and now it read as follows: ‘Thus, we hypothesized that at this growth stage roots did not respond to phosphorus fertilization. A. thaliana is considered a plant that can thrive in marginal soils where optimum nutrient conditions are limited.’ 

L304: These papers might be useful for this discussion on maize need for P early on with respect to yield: https://doi.org/10.1007/s11104-011-0814-y. https://www.nrcresearchpress.com/doi/pdfplus/10.4141/P00-093) Take them or leave them

RESPONSE: Thank you for the recommendation. As suggested, those two relevant papers were included one in the introduction section (Grant et al., 1999) and the second in the discussion section (Nadeem et al., 2011). 

L309: ‘similarity’ do you mean they grouped close together in the PSA? To talk about similarity you would need to use a norm between the treatments.

RESPONSE: Thanks for the observation. We are referring to visual similarity. We have corrected by adding: ‘In the vegetative stage, the root exudates at 50% and 100% phosphorus showed greater visual similarity in the PCA than the exudates at 25% phosphorus, evidencing an initial sensing from the plant in response to its phosphorus demand.’

L323: I think your mention of other functions of exudates is a good explanation and something other authors overlook.

RESPONSE: Thank you. 

346: that ‘the’ shouldn’t be there

RESPONSE: Thank you; ‘the’ was removed.

L351: new paragraph

RESPONSE: Thank you. 

L352: mechanism 1. Your results suggest there is no synergy (in fact the opposite) in terms of P solubilisation by root exudates, see my comment about L255, at least for the 4 acids you picked out. But this doesn’t disprove your mechanism: I think I read somewhere that certain exudates solubilise P from certain soil surfaces, Al oxides, Fe oxides etc (possibly a paper by Gerke about citrate and malate?) Maybe this is why they exude a mixture of acids? Did the soil-analogue you used have this variability in mineral surfaces which soil has?

Nonetheless, I think this deserves some more discussion.

RESPONSE: We addressed this question in comment L255 above.

L357. I like point 2, I had never heard of this before.

RESPONSE: Thank you. 

I don’t think Table S1 is mentioned in the manuscript

RESPONSE: Table S1 show data that is already presented in supplementary figure 3 and 4 in more detail. Therefore, is was deleted it.

---

## [Decision Letter · Decision Letter 1]

8 May 2020

PONE-D-20-04662R1

Role of root exudates on assimilation of phosphorus in young and old Arabidopsis thaliana plants

PLOS ONE

Dear Dr. Vivanco,

Thank you for submitting your manuscript to PLOS ONE. After careful consideration, we feel that it has merit but does not fully meet PLOS ONE’s publication criteria as it currently stands. Therefore, we invite you to submit a revised version of the manuscript that addresses the points raised during the review process.

We are almost there. 

First I apologize for the length of time this step has taken. In addition the reviewers being late, there was a peculiar piece of administrative weirdness that I had to resolve before sending you the decision. That’s all set now. 

As you can see, reviewer 1 is totally satisfied and reviewer 2 raises only a couple of minor points. In fact, reviewer 2 has a very long comment about the ‘additive’ effect but in the revised manuscript, you have downplayed this, so I think most of the comment is moot. You may consider it as you see fit. However, reviewer 2 writes:

“I note the paper “Does the combination of citrate and phytase exudation in Nicotiana tabacum promote the acquisition of endogenous soil organic phosphorus?” which you cite for the synergy of exudates for P solubilisation is an experiment done in soil not NRIB.” Clearly this needs your attention. 

and writes:

“I think when describing the method for the cumulative experiment, you should make it clear whether the mixtures were all  added to 5 mL of liquid NBRIP so that the same amount of P is the same as the individual experiment. Currently it is unclear.”  And this too should be fixed. 

I also read the paper carefully. I uploaded the pdf with my edits. Most of these are places where your English is non-idiomatic or other small problems of style. I also added a few comments in places where I simply could not understand what you meant or where there was some other small problem. Where I understood, I offered a solution. I am not going to repeat them here but please go through them carefully as you revise. 

Two of these comments though I want to bring up here for emphasis. 

This first is the difference between phosphorus and phosphate. You use ‘phosphorus’ almost everywhere but I think this obscures an important difference. In experiments, you are fertilizing the plants with phosphate. I think it is sloppy to talk about adding “1 mM phosphorus” to the plants when you added 1 mM phosphate. Plants take up phosphate, not phosphorus. Likewise, your compounds act to dissolve phosphate. They do not free elemental phosphorus. It seems misleading to describe them as phosphorus-solubilizing compounds. Instead, phosphorus is more of a concept. It is reasonable to write about ‘phosphorus deficiency’ or like that. In some places, the choice is a bit arbitrary. In my edits, I changed ‘phosphorus’ to ‘phosphate’ where the sense was the specific chemical involved, which actually it usually was. 

The second comment concerns figure 3B where you check the solubilizing activity of combinations of the key compounds. The problem here is that you test the single compounds at 7 mM (Fig. 3A). But you test the combinations at that concentration, and also at lower and higher concentrations. I do not see how the higher and lower combinations can be interpreted without also showing values for the single compounds at each concentration. Without that data, I suggest adding the 7 mM combined results as the last bar of figure 3 A and deleting the data for the other combinations (unless you happen to have data for single compounds at the alternate concentrations). By the way, given that the combination solubilized about the same amount of phosphate as did 3-hydroxyproprionate alone, I don’t see how you can even think about an additive effect.

When you submit your revised ms, please do NOT use track changes for any of my edits on the text that you accept with no change. Instead, track only those places where you either do not use my edit at all or you modify it in some way. Also track any changes you make in response to reviewer two. 

All of this should be very straightforward and I expect the next round should go very much faster. 

We would appreciate receiving your revised manuscript by Jun 22 2020 11:59PM. To enhance the reproducibility of your results, we recommend that if applicable you deposit your laboratory protocols in protocols.io, where a protocol can be assigned its own identifier (DOI) such that it can be cited independently in the future. For instructions see: http://journals.plos.org/plosone/s/submission-guidelines#loc-laboratory-protocols

A rebuttal letter that responds to each point raised by the academic editor and reviewer(s). This letter should be uploaded as separate file and labeled 'Response to Reviewers'. The journal requires you to say something but you do not need to go thru all the small things. Just comment on anything where there is some disagreement, etc. A marked-up copy of your manuscript that highlights changes made to the original version. This file should be uploaded as separate file and labeled 'Revised Manuscript with Track Changes'. Please see my note above about how to handle the tracking. An unmarked version of your revised paper without tracked changes. This file should be uploaded as separate file and labeled 'Manuscript'.

We look forward to receiving your revised manuscript.

Kind regards,

Tobias Isaac Baskin

Academic Editor

PLOS ONE

Reviewers' comments:

Reviewer's Responses to Questions

**Comments to the Author**

1. If the authors have adequately addressed your comments raised in a previous round of review and you feel that this manuscript is now acceptable for publication, you may indicate that here to bypass the “Comments to the Author” section, enter your conflict of interest statement in the “Confidential to Editor” section, and submit your "Accept" recommendation.

Reviewer #1: All comments have been addressed

Reviewer #2: (No Response)

2. Is the manuscript technically sound, and do the data support the conclusions?

Reviewer #1: Yes

Reviewer #2: Yes

3. Has the statistical analysis been performed appropriately and rigorously? 

Reviewer #1: I Don't Know

Reviewer #2: Yes

4. Have the authors made all data underlying the findings in their manuscript fully available?

Reviewer #1: Yes

Reviewer #2: Yes

5. Is the manuscript presented in an intelligible fashion and written in standard English?

Reviewer #1: Yes

Reviewer #2: Yes

6. Review Comments to the Author

Reviewer #1: The Ms should now be published in the present form.

One answer to the question the authors made:

Ca- phosphates may account for a high proportion of P in calcareous soils. However even in these soils as well as in other soils often P adsorbed to Fe/Al- surfaces is the oprincipal form of available P. This may be the case because in many soils P desorption from the soil solid phase is konetically and not thermodynamically controlled (see Gerke, 2015 and reference there).

Reviewer #2: Comments to the authors and editor:

Thank you for considering my comments so thoughtfully in the response. I believe the authors have answered the vast majority of my comments, I only have one outstanding query regarding the following in the response document:

'''

L255: The 1.75M is really interesting, 7mM of them mixed does worse than 7mM of any

individually

Again, really interesting results with Fig 4, very hard to interpret though there is lots going on.

RESPONSE: Each compound has a different molarity; therefore we cannot say that the 4 compounds at

1.75 mM add up to 7mM. We conclude that 1) compounds in combination have an additive effect, and

that 2) the addition of one does not inhibit the effect of the other.

'''

I think concluding there is an ‘additive effect’ is a bit miss leading. Here is the standard definition of an additive affect https://www.dictionary.com/browse/additive-effect. The sum of 7mM individually is much more than 7mM combined (I use the 7mmol case now so there is no issues with the molarity. Both cases has 7mM of each.) thus it is not an additive system. This is of course because in the individual cases are done in separate assays and they solubilise the lightly-bound P pool 4 times, while the combined is done in one assay. I think if you want to convince the reader that they are ‘additive’ (and later ‘synergize’ L348 in the discussion) you need to define exactly what it means for the current system to be additive or synergize. Possibly the current experiments and media aren’t suitable to make this conclusion. As I mentioned before, I suspect the compounds ‘synergise’ (I define this to mean to do better than the sum of its parts) when the media has a range of minerals that buffer P and are attacked more efficiently by certain exudates. For example (this is entirely hypothetical) assume the soil is made up of 50% Al-oxide and 50% Fe-oxide and nicotinic acid solubilises P from Al-oxide very effectively but badly from Fe-oxide and malic acid vice-versa. In this case the two compounds would certainly synergise. As I mentioned before I think there is papers about this but I cannot remember, the other reviewer Prof Gerke is likely to know about this and can point you in the right direction. I note the paper “Does the combination of citrate and phytase exudation in Nicotiana tabacum promote the acquisition of endogenous soil organic phosphorus?” which you cite for the synergy of exudates for P solubilisation is an experiment done in soil not NRIB.

I do not think more experiments or analysis is needed as this is a minor result in the overall paper, but a more careful discussion of this is needed in liight of the results.

I do agree with conclusion that the addition of one does not inhibit the other which is a good result in itself.

Detailed comments:

L187 (on tracked document): Should ‘A phosphorus gradient’ here instead be a ‘compound gradient’?

I think when describing the method for the cumulative experiment, you should make it clear whether the mixtures were all added to 5 mL of liquid NBRIP so that the same amount of P is the same as the individual experiment. Currently it is unclear.

I would like to point out to the editor that when responding to one of my comments they also partly answered one of reviewer one’s, this wasn’t clear in the authors response. In particular, reviewers one’s comment:

‘The relevance of the results with respect to P solubilization is questionable since the P form is rather unrepresentative and the exudate quantity is rather high’

Is partly answered by the author’s response to one of my comments in the same vain:

We have added a paragraph that discusses the relevance and ecological significance of organic acids as phosphorus mobilizers and the relevance of the root exudate analysis in liquid cultures. The new paragraph reads as follows: ‘It has been estimated that organic acids constitute 5 to 10 % of the total organic carbon in the soil solution. The concentration of organic anions measured in the soil solution usually range from 100 nM to more than 580 uM in the rhizosphere of cluster roots (Jones, 1998). However, millimolar concentrations of organic anions are likely required in the soil solution to effectively increase soluble P concentration especially in calcareous soils (Strom et al. 2005; Ryan and Jones, 2001). Strom et al. (2005) tested three organic acids (citrate, malic and oxalate) and a wide range of concentrations (1 mM to 100 mM) to evaluate its effects on the mobilization of phosphorus in calcareous soil. The results showed that the phosphorus mobilization of the tested compounds had a low efficiency and its effect varied depending on the type of organic acid, compound concentration, and pH. Further, due to the low phosphorus mobilization efficiency of those compounds it is still argued if the benefit of releasing large amounts of organic acids into the soil will exceed the cost of carbon lost by the plant, which can be seen as an unnecessary trade-off (Strom et al. 2005). However, low efficiency organic acids can be particularly important in phosphorus mobilization for calcareous soils with a limited phosphorus availability for plants. Finally, our evidence supports the above-mentioned hypothesis, that plants release a combination of compounds with different phosphorus-solubilizing efficiencies, at specific stages of growth, to deal with particular phosphorous needs.

7. PLOS authors have the option to publish the peer review history of their article (what does this mean?). If published, this will include your full peer review and any attached files.

Reviewer #1: No

Reviewer #2: No

---

## [Author Response · Author response to Decision Letter 1]

19 May 2020

Reviewers responses

Editor comments:

The second comment concerns figure 3B where you check the solubilizing activity of combinations of the key compounds. The problem here is that you test the single compounds at 7 mM (Fig. 3A). But you test the combinations at that concentration, and also at lower and higher concentrations. I do not see how the higher and lower combinations can be interpreted without also showing values for the single compounds at each concentration. Without that data, I suggest adding the 7 mM combined results as the last bar of figure 3 A and deleting the data for the other combinations (unless you happen to have data for single compounds at the alternate concentrations). By the way, given that the combination solubilized about the same amount of phosphate as did 3-hydroxyproprionate alone, I don’t see how you can even think about an additive effect.

RESPONSE: We agree with the editor. Following your recommendation, the 7 mM combined results from figure 3B was added to figure 3A. Figure 3B was removed. Legend of figure 3 was modified accordingly. Results and methods section were also updated accordingly. 

Reviewer #2: 

Thank you for considering my comments so thoughtfully in the response. I believe the authors have answered the vast majority of my comments, I only have one outstanding query regarding the following in the response document:

'''

L255: The 1.75M is really interesting, 7mM of them mixed does worse than 7mM of any

individually

Again, really interesting results with Fig 4, very hard to interpret though there is lots going on.

RESPONSE: Each compound has a different molarity; therefore we cannot say that the 4 compounds at

1.75 mM add up to 7mM. We conclude that 1) compounds in combination have an additive effect, and

that 2) the addition of one does not inhibit the effect of the other.

'''

I think concluding there is an ‘additive effect’ is a bit miss leading. Here is the standard definition of an additive affect https://www.dictionary.com/browse/additive-effect. The sum of 7mM individually is much more than 7mM combined (I use the 7mmol case now so there is no issues with the molarity. Both cases has 7mM of each.) thus it is not an additive system. This is of course because in the individual cases are done in separate assays and they solubilise the lightly-bound P pool 4 times, while the combined is done in one assay. I think if you want to convince the reader that they are ‘additive’ (and later ‘synergize’ L348 in the discussion) you need to define exactly what it means for the current system to be additive or synergize. Possibly the current experiments and media aren’t suitable to make this conclusion. As I mentioned before, I suspect the compounds ‘synergise’ (I define this to mean to do better than the sum of its parts) when the media has a range of minerals that buffer P and are attacked more efficiently by certain exudates. For example (this is entirely hypothetical) assume the soil is made up of 50% Al-oxide and 50% Fe-oxide and nicotinic acid solubilises P from Al-oxide very effectively but badly from Fe-oxide and malic acid vice-versa. In this case the two compounds would certainly synergise. As I mentioned before I think there is papers about this but I cannot remember, the other reviewer Prof Gerke is likely to know about this and can point you in the right direction. I note the paper “Does the combination of citrate and phytase exudation in Nicotiana tabacum promote the acquisition of endogenous soil organic phosphorus?” which you cite for the synergy of exudates for P solubilisation is an experiment done in soil not NRIB.

I do not think more experiments or analysis is needed as this is a minor result in the overall paper, but a more careful discussion of this is needed in liight of the results.

I do agree with conclusion that the addition of one does not inhibit the other which is a good result in itself.

RESPONSE: We appreciate your follow-up comments. Due to a similar concern by the editor we have modified Figure 3 and made changes to the methods and results sections to address in the best way possible your comment. Please see below: 

1. We incorporated the 7 mM combination from figure 3B as the last column of figure 3A. We removed figure 3B. We agreed that not having single data for every compound with different concentrations, including 1.75 and 28 mM, make difficult to assess the effects. The result (line: 245-254) and method sections (line: 186-193) were modified accordingly to reflect these changes. 

2. With respect to the citation: we were not able to find a similar result under the same conditions used in our study (NBRIP media). We kept the citation because we believe this paper is relevant for our discussion. However, to address your concern, we have clearly stated that the conditions from that finding are different that ours. Now it reads as follow: ‘However, this study was performed under soil conditions and not using liquid NBRIP media”. In addition, it is clearly stated in this paragraph that these are hypotheses that will warrant experimental testing. 

Detailed comments:

L187 (on tracked document): Should ‘A phosphorus gradient’ here instead be a ‘compound gradient’?

RESPONSE: Good observation. The word gradient was replaced by ‘compound mixture’. Now the sentence read as follow: 

‘A compound mixture that included the previous tested concentration (7 mM per compound) was tested in order to compare if the combination of compounds would equal or surpass the effect of a single compound’

I think when describing the method for the cumulative experiment, you should make it clear whether the mixtures were all added to 5 mL of liquid NBRIP so that the same amount of P is the same as the individual experiment. Currently it is unclear.

RESPONSE: Thank you. We added this information to the result section. Now it reads as follow: 

‘…. Briefly, 35 μL (100 mM) of each compound was added to 5 mL liquid NBRIP medium resulting in a final concentration of 7mM. Each compound added to the pool had 7 mM.’

---

## [Editor Report · Decision Letter 2]

21 May 2020

Role of root exudates on assimilation of phosphorus in young and old Arabidopsis thaliana plants

PONE-D-20-04662R2

Dear Dr. Vivanco,

We are pleased to inform you that your manuscript has been judged scientifically suitable for publication and will be formally accepted for publication once it complies with all outstanding technical requirements.

With kind regards,

Tobias Isaac Baskin

Academic Editor

PLOS ONE
---

## [Editor Report · Acceptance letter]

22 May 2020

PONE-D-20-04662R2 

Role of root exudates on assimilation of phosphorus in young and old Arabidopsis thaliana plants 

Dear Dr. Vivanco:

I am pleased to inform you that your manuscript has been deemed suitable for publication in PLOS ONE. Congratulations! Your manuscript is now with our production department. 

With kind regards,

on behalf of

Dr. Tobias Isaac Baskin 

Academic Editor

PLOS ONE